# DCP: Learning Accelerator Dataflow for Neural Network via Propagation

## Abstract

Deep neural network (DNN) hardware (HW) accelerators have achieved great success in improving DNNs' performance and efficiency. One key reason is dataflow in executing a DNN layer, including on-chip data partitioning, computation parallelism, and scheduling policy, which have large impacts on latency and energy consumption. Unlike prior works that required considerable efforts from HW engineers to design suitable dataflows for different DNNs, this work proposes an efficient data-centric approach, named Dataflow Code Propagation (DCP), to automatically find the optimal dataflow for DNN layers in seconds without human effort. It has several attractive benefits that prior arts do not have. (i) We translate the HW dataflow configuration into a code representation in a unified dataflow coding space, which can be optimized by back-propagating gradients given a DNN layer or network. (ii) DCP learns a neural predictor to efficiently update the dataflow codes towards the desired gradient directions to minimize various optimization objectives (e.g., latency and energy). (iii) It can be easily generalized to unseen HW configurations in a zero-shot or few-shot learning manner. For example, without using additional training data, DCP surpasses the GAMMA (Kao & Krishna, 2020) method that performs a full search using thousands of samples. Extensive experiments on several representative models such as MobileNet, ResNet, and ViT show that DCP outperforms its counterparts in various settings.

## 1 Introduction

Deep neural networks (DNNs) have achieved remarkable breakthroughs in many areas, such as vision and language (He et al., 2021), autonomous driving (Al-Qizwini et al., 2017), and biology science (Jumper et al., 2021). However, the exponentially-increased model size often increases the latency and energy consumption of DNN applications, leading to a flourishing area of designing HW-efficient DNN models (Tan et al., 2019; Ma et al., 2018; Iandola et al., 2017) and DNN HW accelerators (Chen et al., 2017; 2019; nvd, 2018; Du et al., 2015). Compared with general-purpose HW processors, DNN accelerators can achieve higher efficiency and lower energy when executing a DNN. This is done by designing a more appropriate micro-architecture and optimizing the DNN's HW mapping strategy, called dataflow, including the order to perform the DNN layer computations and how these computations are mapped across the HW resources (e.g., processing elements and memory). Designing dataflow for optimal on-chip performance and efficiency is a fundamental and challenging task.

The dataflow of existing DNN accelerators is typically over-specialized with under-explored dataflow designs (Chen et al., 2017; nvd, 2018; Du et al., 2015; Jouppi et al., 2017; Parashar et al., 2017; Akhlaghi et al., 2018), hindering the generalization and efficiency of DNN executions. For example, NVDLA (nvd, 2018) exploits parallelism across the input and output channels, while Eyeriss (Chen et al., 2017) exploits parallelism across the filter and input activation rows. We observe that NVDLA is suitable for DNN layers with many channels, while Eyeriss prefers the DNN layers with large filters and feature maps. A quantitative comparison is shown in Fig. 1(a), which compares the energy-delay-product (EDP) of three hand-craft dataflows, i.e., NVDLA, Eyeriss, and ShiDianNao accelerators when processing three different DNN models, including ResNet101 (He et al., 2016), Vision Transformer (Dosovitskiy et al., 2020), and MobileNet-V2 (Sandler et al., 2018). We see that different hand-craft dataflows have their own advantage on certain types of DNN

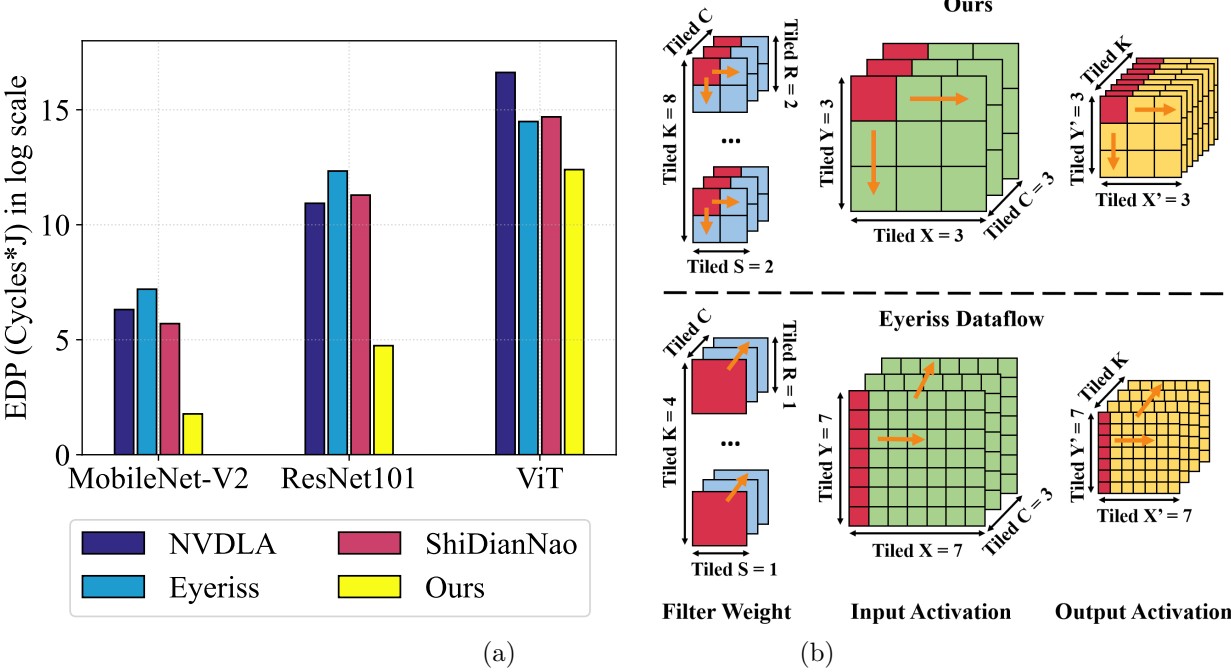

Figure 1: **Dataflow Comparisons and Explanations.** (a) We compare the Energy-delay-product (EDP, lower is better) between NVDLA, Eyeriss, ShiDianNao, and our DCP. We see that DCP can achieve the best efficiency for all three visual models. (b) visualizes and compares the dataflow of ours and Eyeriss using the first layer of ResNet101. The layer dimensions have been tiled based on the partitioning size of dataflow. The red color labels the tiles to perform parallel computation, and the orange arrow implies the computation order, where $K, C, R, S, Y, X, Y', X'$ represent output/input channels, filter row/column, input row/column and output row/column respectively. Our learned dataflow costs only 18.9% read/writes compared to Eyeriss by (i) using a smaller kernel tiled size ($4 \times 4$ versus $1 \times 1$ in Eyeriss) and smaller tiled output channels (8 versus 4 in Eyeriss) but a larger one for input ($3 \times 3$ versus $7 \times 7$ in Eyeriss) and (ii) different computation order of dimensions.

layers. Customizing dataflows for different DNNs is more flexible and efficient than a fixed one for all DNNs. Although reconfigurable accelerators (Chen et al., 2019; Kwon et al., 2018) have extra circuits to enable a configurable dataflow, which can be tuned at compile time to leverage the HW resources fully, the manually-designed dataflows still dominate in existing accelerators. To alleviate manual efforts in designing dataflows, recent works (Yazdanbakhsh et al., 2021; Kao & Krishna, 2020; Kao et al., 2020; Wang et al., 2021) have studied autonomous algorithms for dataflow design using some conventional search techniques such as exhaustive search (Wang et al., 2021; Parashar et al., 2019) and reinforcement learning (Kao et al., 2020). However, it is hard for these methods to deal with a vast dataflow design space, e.g., $O(10^{36})$ for one DNN layer, as discussed in Sec. 4.1. To cope with this challenge, a straightforward strategy (Yazdanbakhsh et al., 2021; Kao et al., 2020; Wang et al., 2021) is to scale down the design space, leading to sub-optimal design.

This paper answers a question naturally raised from the above issues: *can we efficiently generate optimal dataflows for different DNN layers and networks?* To efficiently search an optimal dataflow in a vast design space, we propose Dataflow Code Propagation (DCP), an autonomous pipeline to search dataflow for many DNN applications. Unlike prior work, DCP uses an encoding scheme to translate the dataflow and DNN layer configuration into a unified coding representation and build a vast coding space of dataflow design. By leveraging this encoding scheme, we build a benchmark with the input of unified codes and the output of corresponding evaluation metrics (e.g., latency, energy, etc.). We then learn a neural predictor to project the unified code into the corresponding evaluation metrics. By optimizing the evaluation metrics, we back-propagate the gradient of the neural predictor and update the unified code directly along the gradients of various optimization objectives, such as more negligible latency and energy.

Compared with prior dataflow designing techniques (Kao & Krishna, 2020; Parashar et al., 2019; Yang et al., 2020b), our proposed DCP has several appealing properties. Firstly, DCP is data-efficient in the sense that it can search the optimal dataflow for various DNNs in seconds once the neural predictor is trained. But previous works need to search dataflow for each DNN with a time-consuming simulation process. Secondly, it can optimize dataflow optimization at the whole model level by accumulating gradients of all layers. Fig. 1(b) visualizes the searched dataflow by DCP, which outperforms many manually-designed dataflows such as Eyeriss (Chen et al., 2017). Lastly, DCP is easily extended to the multi-objective dataflow optimization, yielding the dataflow configuration that can trade off multiple HW metrics better.

We make three main **contributions**. (i) We propose an encoding scheme by translating the DNN layer configurations and accompanying dataflows into a coding representation by presenting a unified coding space and a comprehensive dataflow benchmark. (ii) We back-propagate the gradient of the neural predictor to efficiently update dataflow codes in the vast design space to achieve the desired optimization goals. To our knowledge, this is the first work that leverages differentiable back-propagation in dataflow optimization. (iii) Extensive experiments show the generalization of DCP by customizing dataflow for many optimization objectives (e.g., latency/energy of single/multiple layers) in various DNNs in both seen and unseen HW configurations.

## 2 Related work

**DNN Accelerators.** The development of DNNs has led to the flourishing of specialized DNN accelerators in recent years. According to the flexibility of DNN accelerators' dataflow, DNN accelerators can be categorized into two categories which are fixed dataflow accelerators (Chen et al., 2017; nvd, 2018; Du et al., 2015) and reconfigurable accelerators (Chen et al., 2019; Kwon et al., 2018). Application-specific accelerators (Kawamoto et al., 2020a; Yamada et al., 2019; Kawamoto et al., 2020b) design a dataflow fixed and optimized for one or several DNN applications. Reconfigurable accelerators have extra circuits to enable a configurable dataflow, which can be tuned at compile time. Therefore, an exemplary dataflow can not only guide the architecture design for the DNN accelerators with a dataflow baked into the silicon but also let the DNN applications can fully leverage the HW resources of the DNN accelerators with a configurable dataflow. The optimization of dataflow is central in designing both fixed dataflow accelerators and reconfigurable accelerators. And in this work, we aim to optimize the dataflow design for various DNN models, which is central to both fixed dataflow accelerators and reconfigurable accelerators.

**Design Space Exploration of DNN Accelerator.** In the design of a DNN accelerator, several components should be considered. These components also form the design space of the DNN accelerator, which are dataflow, HW resources, specific circuit design, and so on. For designing efficient specialized DNN accelerators for target applications, many works (Yazdanbakhsh et al., 2021; Kao et al., 2020; Wang et al., 2021) have been proposed to explore the design space of DNN accelerators. For instance, Apollo (Yazdanbakhsh et al., 2021) is a framework for optimizing the DNN accelerator based on the features extracted with Integrated Circuit (IC) design knowledge. Our work can also broadly be categorized into the design space exploration algorithm of the DNN accelerator, as dataflow is an essential component in the DNN accelerator's design space.

**Performance Simulator of DNN Accelerator.** Using Electronic Design Automation (EDA) tools to evaluate the performance of the DNN accelerator requires the corresponding circuit design, and the simulation process is also time-consuming. Therefore, to efficiently explore the design space of the DNN accelerator, several performance simulators (Kwon et al., 2019; Parashar et al., 2019; Yang et al., 2020b; Wang et al., 2021) have been proposed to provide accurate performance simulation for possible dataflows and accompanying HW resource configurations. According to the expression of data reuse, most simulators are compute-centric, which uses the loop-nest representation to infer data reuse. In this paper, we choose MAESTRO (Kwon et al., 2019), a data-centric simulator that uses spatial and temporal maps to express data reuse.

**Co-design of Network and DNN Accelerator.** Except for the growing work exploring the design space of network architecture or DNN accelerators, co-design the DNNs and their accelerator have attracted increasing research interests in recent years. Some of them merge the design space of the network and DNN accelerator and search the optimal network-accelerator pair jointly with the method of evolutionary algorithm (Fasfous

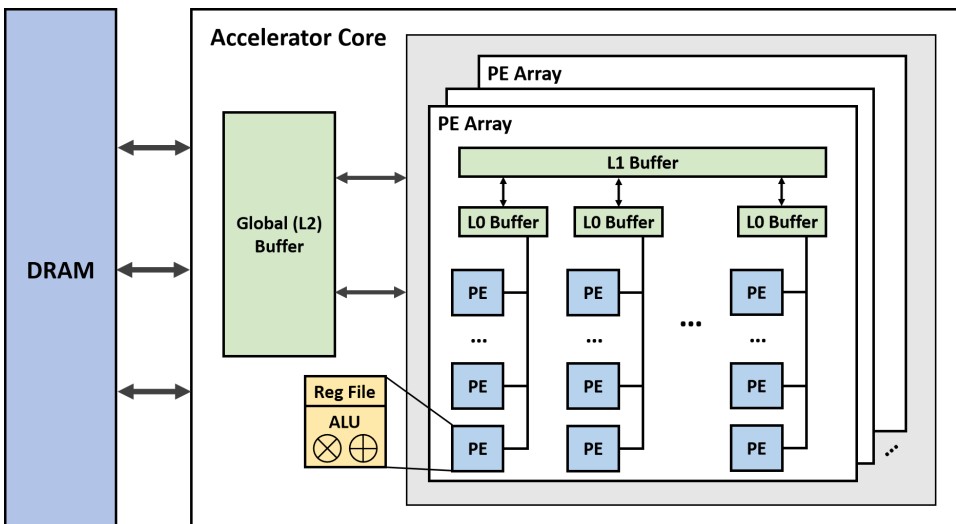

Figure 2: **Example of DNN Accelerator.** An abstract DNN accelerator architecture that contains a hierarchical memory system and PE arrays. This abstract DNN accelerator architecture is also used in many DNN accelerators (Chen et al., 2017; Jouppi et al., 2017; Parashar et al., 2017; Akhlaghi et al., 2018).

et al., 2022; Lin et al., 2021) or gradient-based methods (Fasfous et al., 2021; Choi et al., 2021; Li et al., 2020). While some methods (Zhou et al., 2022; Abdelfattah et al., 2020) tackle this co-design problem with the manner of reinforcement learning. However, these works target different HW platforms, including academic accelerators (Yang et al., 2020a), performance simulators (Lin et al., 2021; Choi et al., 2021), and commercial accelerators (Zhou et al., 2022). The differences in target hardware platforms also introduce difficulties in the performance comparison as all the result posted in related papers needs to be performed from scratch for different hardware platforms. Another aporia is the modeling method of the hardware accelerator is also different in many prior works, which makes it hard to compare with the work targeting other hardware platforms. And different from those works in co-designing DNN models and HW configurations, our DCP focuses on optimizing dataflow design for various DNN models. By building the connection between HW metrics and dataflow using a neural predictor, DCP can achieve better HW performance than NAAS (Lin et al., 2021).

## 3 Preliminary

**DNN Accelerators.** DNN accelerators have HW architectures specifically designed to run the DNN applications efficiently. To better understand the architecture of DNN accelerators, an abstract DNN accelerator architecture used in many state-of-the-art accelerators is shown in Fig. 2. As illustrated in Fig. 2, most DNN accelerators comprise several arrays of Processing Elements (PEs) to exploit the parallelization and data reuse opportunities in DNN applications. Typically, a PE will have a specific arithmetic logical unit (ALU) to perform the multiply-accumulate operations (MACs) and a local register file to store the input and output of the ALU. Within the PE array, there will have a local scratchpad memory ("L1 Buffer") to assign the data to PEs. Sometimes, there will be another local scratchpad memory ("L0 Buffer") between L1 Buffer and PEs if the accelerator wants to do more fine-grained control with PEs. In addition to the L1 Buffer, most DNN accelerators also have a global scratchpad memory ("L2 buffer") shared among PE arrays to stage data to feed PEs for reducing the number of energy-expensive and time-consuming accesses of dynamic random access memory (DRAM).

**Dataflow.** Dataflow is the data communication pattern within a DNN accelerator between the compute and memory elements. Dataflow affects the data reuse pattern and parallelism strategy of the accelerator. As it has been shown that the energy cost of moving data exceeds the cost of computation (Horowitz, 2014), understanding and optimizing dataflow is a critical component of DNN accelerator design, as it directly determines how data is transferred between different levels of buffers in the memory hierarchy. Specifically,

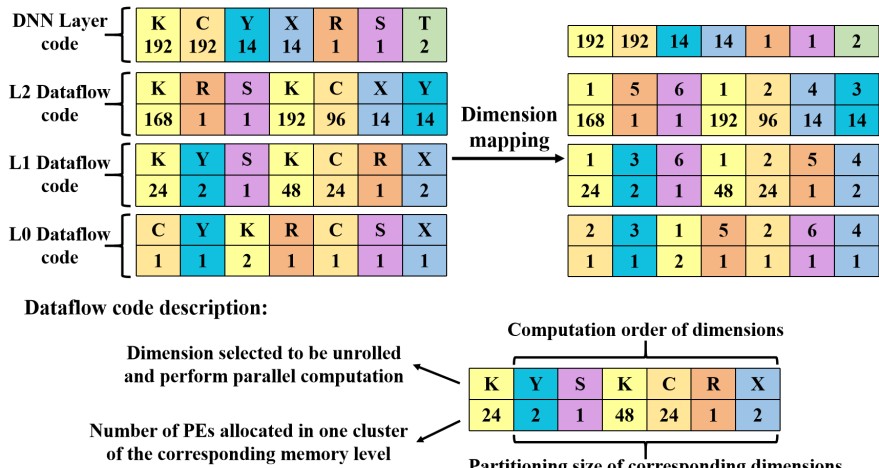

Figure 3: **Coding representations of dataflow and DNN layer.** DNN layer code is a seven-dimensional code that describes the DNN layer in the dimensions of $K, C, Y, X, R, S, T$. We use a dataflow optimized for MobileNet-v2 as an example. The dataflow code describes the three memory levels of the accelerator. The L1 dataflow code is an example, the first line is the index of seven dimensions, and the second line is their accompanying numbers. These seven dimensions contain six dimensions in the DNN layer code except T in computation order and a parallel dimension selected from them to perform parallel computation. The accompanying number of the parallel dimension specifies the number of PEs allocated in one cluster of L1 memory. In contrast, the partitioning size is the accompanying number of the rest six dimensions.

we describe dataflow in three aspects: parallelism, computation order, and partitioning size (see more details in Appendix Sec.A).

- **Parallelism.** Parallelism is about allocating the computation workload into the multiple HW processing units and letting these processing units can perform the computation simultaneously. For DNN accelerators, the processing units are PEs, and they usually achieve parallelism via unrolling the loop nest of convolutional layers spatially.

- **Computation Order.** Computation order is the temporal order of the dimensions to be performed in the multiple-dimensional computation task. Different computation orders can exploit different data movement patterns and reuse opportunities.

- **Partitioning Size.** As there are many parameters to the DNN application and the buffer size of DNN accelerators is limited, accelerators with a multi-level memory hierarchy will partition these parameters and allocate them into buffers. In this process, partitioning size will determine the data size of each tensor (input/output/weight) that needs to be present within the accelerator's buffer.

Although different types of DNN prefer different dataflow designs, as indicated in Fig. 1, existing accelerators either utilize manually-designed dataflows Chen et al. (2017); nvd (2018); Du et al. (2015) or search dataflow using conventional optimization algorithms Wang et al. (2021); Yang et al. (2020b); Parashar et al. (2019), leading to serious human efforts and sub-optimal performance. To tackle this problem, we propose a data-centric method to find the optimal dataflow for DNN layers in seconds without human effort.

## 4 Our Approach

In this section, we propose Dataflow Code Propagation (DCP) to efficiently search the optimal dataflow for DNN accelerators in a differentiable manner. Towards this goal, we first benchmark the dataflow parameters and DNN layer configurations in Sec.4.1. Then, a predictor is trained to predict HW metrics given the dataflow and DNN layer parameters in Sec.4.2. Lastly, we search the optimal dataflow for DNN layers by back-propagating the gradients under various objectives such as minimizing latency consumption in Sec.4.3. An overview of DCP is shown in Fig. 4. In addition, leveraging its general pipeline, DCP can easily handle

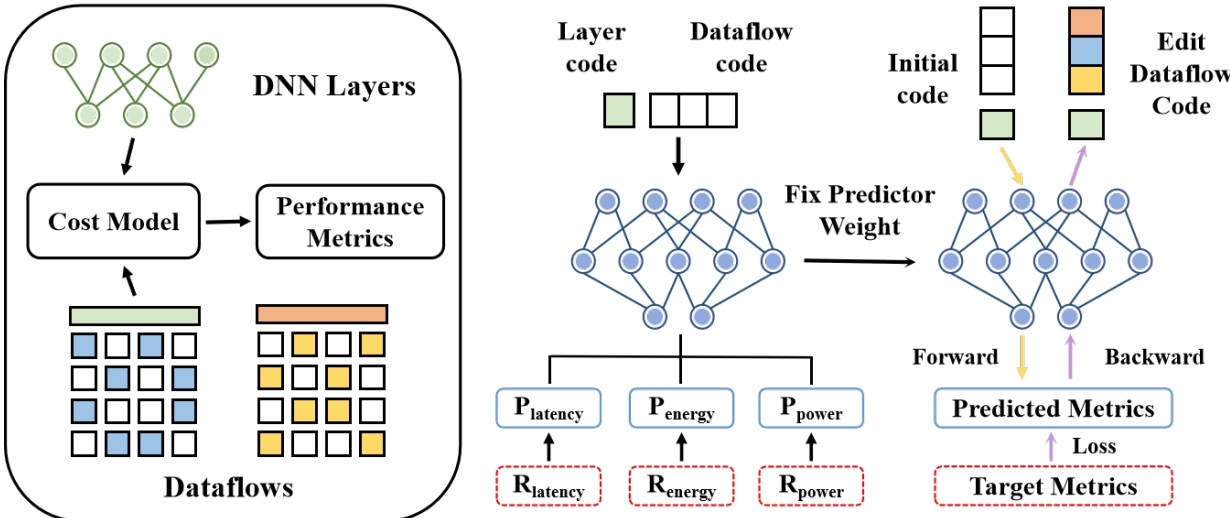

Figure 4: **An overview of our Dataflow Code Propagation (DCP).** DCP first builds a benchmark of dataflow and DNN layer and then trains a predictor to predict various HW metrics. Finally, we back-propagate the gradients of the fixed neural predictor to efficiently update the dataflow code towards the optimization objective (lower predicted metrics), conditioned on the layer code.

unseen hardware constraints and be applied to other hardware models, which is illustrated in Sec.5.5 and Appendix. B respectively.

In addition, DCP can easily handle unseen hardware constraints. Note that the benchmark of dataflow and DNN layer is built under the default HW setting of the Eyeriss chip (i.e. 168 PEs and 108KB on-chip memory). In Sec. 5.5, we show that the neural predictor trained on such a benchmark can be easily transferred to other benchmarks built under unseen HW settings with few-shot/zero-shot learning pipelines. Moreover, we would like to highlight that the overall pipeline of DCP (i.e. build benchmark –>train predictor – >search dataflow) is general enough to be applied to other hardware models, which is demonstrated by the Space-Time-Transformation matrix used in Tensorlib in Appendix B.

### 4.1 Benchmarking Dataflow and DNN Layer

We propose an encoding scheme to build the benchmark of the dataflow and the DNN Layer where the HW metrics for each pair of dataflow and the DNN Layer are evaluated by HW simulation.

**Encoding Scheme.** We encode the dataflow and accompanying DNN layer configurations into coding representations to perform the dataflow optimization efficiently. A detailed example of our coding representation is shown in Fig. 3. In our encoding scheme, the configuration of a DNN layer is represented by seven dimensions, which are input row/column ($Y/X$), filter row/column ($R/S$), output/input channels ($K/C$), and an extra dimension ($T$) to describe the layer type of DNN layers. The row/column of output can be deduced by the input row/column and filter row/column. Overall, we encode a DNN layer into a seven-dimensional code in the order of $K, C, Y, X, R, S, T$, denoted by $x \in \mathbb{R}^m$ where $m = 7$.

For the coding representation of dataflow, we encode it into a $(3, 7, 2)$ dimensional code, denoted by $y \in \mathbb{R}^n$ where $n = 42$. Considering the architecture of the DNN accelerator introduced in Sec. 3, this '3' refers to the three memory levels of an accelerator. For each memory level, there will be a $(7, 2)$ dimensional code to describe the dataflow of one cluster in the corresponding memory level. As demonstrated in Fig. 2, a DNN accelerator may have one L2 cluster and several L1/L0 clusters, which contain a memory buffer and corresponding sub-clusters. The '7' in the $(7, 2)$ dimensional code of each memory level addressed the seven dimensions. Notably, the first dimension indicates a parallel dimension selected from $K, C, Y, X, R, S$ to be unrolled and perform parallel computation. The rest six dimensions are obtained by reordering the dimensions of $K, C, Y, X, R, S$ with computation order. Then, the two-dimensional code '2' of each dimension contains a

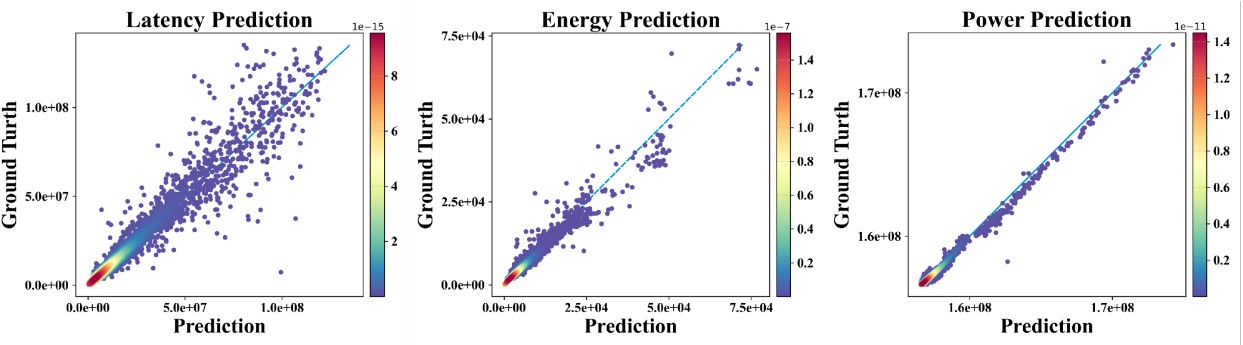

Figure 5: Regression performance of the neural predictor for the HW metrics of latency, energy, and power, respectively. The color bar represents the density of data points calculated by Gaussian kernel density estimation (KDE). The darker the color is, the more concentrated the data points distribute.

number to index the corresponding dimension and an accompanying number. The accompanying number of the parallel dimension specifies how many PEs will be allocated to perform parallel computation. In contrast, the accompanying number of the rest six dimensions determines their partitioning size. The size of memory buffers in each sub-clusters is calculated by partitioning size, as the buffers store input, output, and weight tensors. In contrast, the size of these tensors is the linear combination of partitioning sizes.

**Hardware Metrics.** To build the benchmark, we evaluate the HW metrics for each pair of code representations of the dataflow and the DNN Layer. However, the above encoding scheme involves a vast search space, making it hard to enumerate all DNN layer dimensions and their corresponding dataflows. For example, the first layer in ResNet101 ($K = 64, C = 3, Y = 224, X = 224, R = 7, S = 7$) can form a dataflow search space with the complexity of $O(10^{36}) = (64 \times 3 \times 224^2 \times 72 \times 6! \times 6)^3$. In practice, we randomly sample $2K$ DNN layers based on all classification models in the torchvision (tor, 2022) library (version 0.12, including ViT). For each DNN layer, we sample $2K$ dataflows randomly. We find that such a scale of the dataset is enough to train a good predictor. Then, we evaluate the performance for all sampled dataflows and accompanying DNN layers with a performance simulator (Kwon et al., 2019) to form a dataset with the input of our unified coding representation and the output of performance metrics (latency, energy, etc.). An extensive experiment based on cycle-accurate simulation is shown in Appendix.B.

**Collection Data with HW Constraints.** In the design of realistic DNN accelerators, several constraints should be considered, like the number of PEs contained in the accelerator and the memory size of each buffer. The dataflow encoding scheme introduced above can also address the design constraints of realistic DNN accelerators. The simple one is that the accompanying number of parallels can constrain the number of processing elements in the accelerator and corresponding sub-clusters. We address these constraints of accelerator design by only sampling data satisfying the limitation of these constraints. In specific, we follow the same hardware constraints as it is presented in Eyeriss (Chen et al., 2017) chip, which consists of i) The number of PE cannot surpass 168. ii) The on-chip memory capacity cannot exceed 108 KB. iii) The number of PE of a sub-cluster cannot exceed its parent cluster. iv) The memory capacity of a sub-cluster cannot exceed its parent cluster.

## 4.2 Training Predictor

Unlike previous dataflow designing approaches that require a time-consuming HW simulation process for each model, DCP aims to train a predictor using the benchmark introduced in Sec.4.1 to optimize dataflow for all models, as shown in Fig.4.

**Predictor Architecture.** The neural predictor $\mathcal{F}$ is instantiated by an encoder $\theta_e$ and several prediction heads $\{\theta_i\}$ for HW metrics where $\theta_i$ indicates the head $i$-th metric. HW metrics use different prediction heads but share a common encoder can not only encourage the predictor to predict the HW metrics well but also reduce computations and parameters. The encoder is a shallow attention network with a 4-layer self-attention layer (Subakan et al., 2021) of hidden size 64. The activation function is Swish (Ramachandran

Table 1: Performance of optimized dataflow for different optimization methods on several DNN models. Each method is performed multiple times with different optimization objectives.

| Method | MobileNet-V2 (Sandler et al., 2018) | | | ResNet101 (He et al., 2016) | | | ViT (Dosovitskiy et al., 2020) | | | Time cost |
|---|---|---|---|---|---|---|---|---|---|---|
| | EDP (cycles * nJ) | Latency (cycles) | Energy (nJ) | EDP (cycles * nJ) | Latency (cycles) | Energy (nJ) | EDP (cycles * nJ) | Latency (cycles) | Energy (nJ) | |
| PSO (Marini & Walczak, 2015) | 2.14e+10 | 1.72e+07 | 2.20e+05 | 5.05e+11 | 4.79e+07 | 5.90e+05 | 1.72e+13 | 1.58e+08 | 8.52e+05 | 4945s |
| Portfolio (McMillan et al., 2011) | 1.75e+10 | 1.64e+07 | 2.16e+05 | 1.78e+11 | 1.51e+07 | 5.45e+05 | 2.75e+12 | 5.50e+06 | 6.38e+05 | 6010s |
| OnePlusOne (Igel et al., 2006) | 4.44e+10 | 2.14e+07 | 2.37e+05 | 4.11e+12 | 1.41e+08 | 6.86e+05 | 2.56e+15 | 3.08e+08 | 8.86e+05 | 4756s |
| CMA (Hansen et al., 2009) | 7.83e+11 | 2.91e+07 | 2.64e+05 | 2.77e+19 | 9.22e+18 | 9.04e+05 | 3.14e+12 | 1.79e+07 | 1.24e+06 | 7529s |
| DE (Vodopija et al., 2018) | 1.78e+10 | 1.64e+07 | 2.11e+05 | 9.14e+10 | 1.51e+07 | 5.20e+05 | 3.09e+11 | 4.79e+05 | 6.84e+05 | 4882s |
| TBPSA (Poláková et al., 2017) | 1.90e+10 | 1.68e+07 | 2.17e+05 | 2.90e+11 | 2.49e+07 | 5.47e+05 | 4.01e+11 | 8.18e+06 | 6.35e+05 | 4781s |
| pureGA (Goldberg, 1989) | 3.91e+10 | 2.44e+07 | 2.20e+05 | 1.64e+12 | 1.17e+08 | 6.13e+05 | 4.08e+12 | 4.64e+06 | 7.91e+05 | 6662s |
| Random (Bergstra & Bengio, 2012) | 5.12e+10 | 2.08e+07 | 2.19e+05 | 1.70e+12 | 1.16e+08 | 5.94e+05 | 5.16e+11 | 3.59e+06 | 8.30e+05 | 4403s |
| GAMMA (Kao & Krishna, 2020) | 2.11e+09 | 7.63e+05 | 1.14e+05 | 2.31e+11 | 1.19e+07 | 6.62e+05 | 9.67e+09 | 2.95e+05 | 8.56e+05 | 5253s |
| DCP (Ours) | **8.88e+06** | **9.66e+04** | **3.51e+03** | **7.18e+07** | **4.64e+05** | **1.01e+04** | **5.70e+07** | **1.31e+05** | **1.72e+04** | **3606s** |

et al., 2017), and a dropout (Srivastava et al., 2014) of 0.1 is also applied to mitigate the over-fitting of each layer. Note that a linear projection layer is used to project the input to the feature with the size of 512. It is then reshaped to the size of $8 \times 64$ where '8' is the token length and '64' is the hidden size. Moreover, each prediction head is instantiated by a single attention layer with an output size of 1. In total, this projector has $0.4M$ parameters and $1M$ FLOPs. Hence, the training cost of the predictor is almost negligible.

**Training Details.** The loss function for training the predictor is generated by three HW metrics (i.e., latency, energy, and power), as shown in Fig.4, considering that both running time and energy consumption are critical for efficient dataflow design. Let $\mathcal{G}$ denote the loss function. The training loss $\mathcal{L}$ is given by

$$\mathcal{L} = \sum_{i=1}^{M} \mathcal{G}(P_i, R_i) \text{ and } P_i = \mathcal{F}(\theta_e, \theta_i; [x, y]) \tag{1}$$

where $M$ is the number of HW metrics and $M = 3$ in our case. $P_i$ and $R_i$ are log-scaled predicted and ground-truth values of the $i$-th metric. $x$ and $y$ are the code representations of the DNN layer and accelerator dataflow respectively. In implementation, $\mathcal{G}$ is log-cosh function as written by $\mathcal{G}(P, R) = \log(\cosh(P - R))$.

The predictor is trained on the benchmark in Sec. 4.1 where 80% and 20% data are used as training and validation sets. In addition, we use Adam optimizer (Kingma & Ba, 2015) with parameter $\beta = (0.9, 0.999)$ and $\epsilon = 1e - 3$. The predictor is trained for 50 epochs with initial learning $1e - 2$ and weight decay $1e - 7$.

The regression performance of our neural predictor is shown in Fig. 5. It can be seen that the predictor can predict various metrics given dataflow and the DNN layer.

## 4.3 Dataflow Code Optimization

After training the neural predictor, the dataflow optimization becomes ultra-fast by back-propagating along the direction of target metrics' gradients. As shown in Fig.4, the predictor will forward an initial dataflow code to obtain predicted performance metrics $P_i$. To minimize (maximize) a metric, we set the target value of the metric as $R'_i = P_i - 1$ ($R'_i = P_i + 1$). Then the weight of the predictor will be fixed to perform the back-propagation with respect to the code of dataflow, returning an optimized dataflow after a certain number of iterations. We investigate optimizing dataflow from layer-level, model-level, and multi-objective perspectives.

**Layer-level Dataflow Propagation.** The layer-level dataflow propagation aims to search optimal dataflow for each layer in a given deep model. Let $\{x_l\}_{l=1}^{L}$ be the code representations of all layers in the model where $L$ is the number of layers. The layer-level dataflow optimization is derived by gradient descent. For all $l \in [L]$,

$$y_l^{t+1} = y_l^t - \eta \frac{\partial \mathcal{G}(\mathcal{F}(\theta_e^{\text{fix}}, \theta_i^{\text{fix}}; [x_l, y]), R'_i)}{\partial y}\Big|_{y_l^t} \tag{2}$$

where $t = 0, 1 \cdots, T - 1$ is the iteration index, $\eta$ is the stepsize, $\theta_e^{\text{fix}}$ and $\theta_i^{\text{fix}}$ represent fixed encoder and prediction head of $i$-th metric, respectively. Note that Eqn.(2) cannot guarantee that the searched dataflow satisfies the HW constraints, such as the integer requirement of memory. Hence, two modifications are adopted. Firstly, we round the searched dataflow every $T_{int}$ iteration to obtain dataflow parameters with the

Table 2: Dataflow performance of DCP, NAAS, and a suite of DNN accelerators with fixed dataflow. DCP's dataflow is searched by model dataflow propagation, and it is fixed for every DNN model.

| Method | MobileNet-V2 (Sandler et al., 2018) | | | ResNet101 (He et al., 2016) | | | ViT (Dosovitskiy et al., 2020) | | |
|---|---|---|---|---|---|---|---|---|---|
| | EDP (cycles * nJ) | Latency (cycles) | Energy (nJ) | EDP (cycles * nJ) | Latency (cycles) | Energy (nJ) | EDP (cycles * nJ) | Latency (cycles) | Energy (nJ) |
| NVDLA (nvd, 2018) | 5.51e+11 | 1.46e+07 | 1.15e+06 | 5.60e+13 | 1.84e+08 | 2.94e+07 | 1.66e+16 | 4.31e+09 | 4.51e+07 |
| Eyeriss (Chen et al., 2017) | 1.34e+12 | 2.33e+07 | 1.71e+06 | 2.28e+14 | 3.92e+08 | 3.93e+07 | 1.96e+15 | 2.53e+08 | 5.37e+07 |
| ShiDianNao (Du et al., 2015) | 2.99e+11 | 5.26e+06 | 1.85e+06 | 8.00e+13 | 1.20e+08 | 4.48e+07 | 2.40e+15 | 1.73e+08 | 5.71e+07 |
| NAAS (Lin et al., 2021) | 1.09e+10 | 9.23e+05 | 3.85e+05 | 5.56e+10 | 2.67e+07 | 1.39e+06 | - | - | - |
| DCP (Ours) | **6.64e+08** | **7.12e+05** | **1.19e+04** | **5.96e+09** | **9.96e+06** | **5.12e+04** | **6.33e+09** | **2.12e+06** | **6.55e+04** |

'-' denotes the corresponding data are not mentioned in the relevant paper

Table 3: DCP's performance of dataflow optimized for single and multiple objectives. Multi-objective optimization includes the pairwise combination of all objectives in single-objective optimization.

| Target | | Single Objective | | | Multiple Objective | | |
|---|---|---|---|---|---|---|---|
| | | Latency | Energy | EDP | Latency + Energy | Latency + EDP | Energy + EDP |
| MobileNet-V2 (Sandler et al., 2018) | Latency (cycles) | 9.66e+04 | 3.91e+05 | 9.83e+04 | 1.04e+05 | 1.10e+05 | 2.21e+05 |
| | Energy (nJ) | 5.49e+03 | 3.51e+03 | 4.69e+03 | 4.85e+03 | 4.75e+03 | 3.88e+03 |
| | EDP (cycles * nJ) | 1.02e+07 | 2.77e+07 | 8.88e+06 | 1.05e+07 | 1.11e+07 | 1.92e+07 |
| ResNet101 (He et al., 2016) | Latency (cycles) | 4.64e+05 | 2.20e+06 | 5.01e+05 | 5.29e+05 | 5.28e+05 | 1.31e+06 |
| | Energy (nJ) | 1.76e+04 | 1.01e+04 | 1.41e+04 | 1.45e+04 | 1.41e+04 | 1.08e+04 |
| | EDP (cycles * nJ) | 7.88e+7 | 3.00e+08 | 7.18e+07 | 8.26e+07 | 8.16e+07 | 1.94e+08 |
| ViT (Dosovitskiy et al., 2020) | Latency (cycles) | 1.31e+05 | 8.76e+05 | 1.39e+05 | 1.76e+05 | 1.73e+05 | 1.88e+05 |
| | Energy (nJ) | 2.80e+04 | 1.72e+04 | 1.88e+04 | 2.07e+04 | 2.18e+04 | 1.99e+04 |
| | EDP (cycles * nJ) | 7.28e+07 | 3.39e+08 | 5.70e+07 | 8.24e+07 | 8.67e+07 | 8.59e+07 |

integer. Secondly, a min-max clip strategy is used for the dataflow code of every sub-cluster to ensure that its parameters do not exceed the value of its parent cluster and also limit the overall HW resource within the constraints of Eyeriss (Chen et al., 2017) Chip.

**Model-level Dataflow Propagation.** Although the layer-level dataflow propagation is optimal, it can only be implemented when dataflow is configurable in an HW accelerator. To alleviate this limitation, we also perform the model-level dataflow propagation by accumulating the propagation gradient of all layers within the target DNN model:

$$y^{t+1} = y^t - \eta \sum_{l=1}^{L} \frac{\partial \mathcal{G}(\mathcal{F}(\theta_e^{\text{fix}}, \theta_i^{\text{fix}}; [x_l, y]), R_i')}{\partial y}|_{y^t}. \tag{3}$$

The summation means that the layer with a larger number would affect the dataflow update more than the smaller ones. By Eqn.(3), DCP can search for the best dataflow that is optimized for all layers within a DNN model simultaneously.

**Multiple Objective Propagation.** In HW design, we often need to trade-off between latency and energy consumption. To this end, we also use multiple objective propagations to search dataflow, which can be expressed by:

$$y_l^{t+1} = y_l^t - \eta \sum_{i=1}^{M} \lambda_i \frac{\partial \mathcal{G}(\mathcal{F}(\theta_e^{\text{fix}}, \theta_i^{\text{fix}}; [x_l, y]), R_i')}{\partial y}|_{y_l^t}. \tag{4}$$

where $\sum_{i=1}^{M} \lambda_i = 1$ uses the weighted average of the gradients of target metrics to perform the back-propagation. Since DCP is fast enough, we can obtain optimal $\{\lambda_i\}_{i=1}^{M}$ by grid search. The summation of gradients in Eqn.(4) helps find dataflow configurations that can trade off multiple HW metrics better. As DCP is a search method based on the neural predictor, we can leverage the power of GPU to perform the grid search of lambda. And the total time cost of the grid search for one multi-objective combination is around 75 seconds.

# 5 Experiments

In this section, we first present our experiment settings in Sec.5.1. Then, we evaluate our proposed DCP technique in dataflow customization from the perspective of per-layer optimization in Sec.5.2, per-model optimization in Sec.5.3 and multi-objectives optimization in Sec.5.4. Lastly, optimizations for unseen HW settings are also performed to show the generalization of DCP in Sec.5.5. Besides the aforementioned experiment, an investigation of the optimization for a realistic HW platform and an ablation study for the design decision of DCP are shown in Appendix.B and Appendix.5.6 respectively.

## 5.1 Experiment Settings

We consider three representative DNN models with different topology architectures and complexity, i.e. MobileNet-V2 (Sandler et al., 2018), ResNet101 (He et al., 2016), and Vision Transformer (ViT) (Dosovitskiy et al., 2020). We evaluate the following optimization methods as the baseline. (i) Random Search: The random search samples the design points randomly and keeps the best solutions. (ii) Genetic Algorithms (GA): In this baseline, we consider a general GA (Goldberg, 1989) and GAMMA (Kao & Krishna, 2020), a specifically designed GA for optimizing the dataflow. (iii) Nevergrad: The rest of our baseline methods are implemented in an optimization algorithm platform Nevergrad (Rapin & Teytaud, 2018), including Particle Swarm Optimization (PSO) (Marini & Walczak, 2015), Passive Portfolio (McMillan et al., 2011), $(1 + 1)$ Evolution Strategy (OnePlusOne) (Igel et al., 2006), Covariance Matrix Adaptation-ES (CMA) (Hansen et al., 2009), Differential Evolution (DE) (Vodopija et al., 2018), and Test-based Population-Size Adaptation (TBPSA) (Poláková et al., 2017).

## 5.2 Per-layer dataflow optimization

We first perform the per-layer dataflow optimization to evaluate DCP's performance in searching for the optimal dataflow for each DNN layer within a model and guide the dataflow design of reconfigurable accelerators. In per-layer optimization, DCP and the baseline methods search the optimized dataflow for every layer within a DNN model. All optimization methods in our evaluation are performed multiple times with different objectives (EDP, latency, and energy), and the detailed performance is summarized in Table 1.

**Comparison of HW Metrics.** In Table 1, we chose EDP as the target objective because it is a well-established metric in the literature like NAAS (Lin et al., 2021), and we aimed to ensure that our results could be compared to those of other state-of-the-art approaches. We set $10K$ samples as the searching constraint of baseline methods, as we observed that the baseline methods could converge on most DNN Layers after evaluating $10K$ samples. However, for every selected model, some methods are hard to find any good solutions within $10K$ trials. This observation also reflects the data efficiency of DCP as our neural predictor is trained on a dataset with $2K$ samples for each DNN layer and does not need to evaluate any more samples in back-propagation. For the optimization performance, DCP outperforms all the baseline methods with *several orders of magnitude* in various HW metrics of MobileNet-V2, ResNet101, and ViT.

**Comparison of Time Cost.** We further compare the time cost of DCP and other baseline methods in searching the optimal dataflow for all evaluation metrics of all three selected DNN models. As baseline methods perform dataflow searching and evaluate it with the cost model simultaneously, the time cost of DCP is inclusive of three parts, which are dataset collection, training predictor, and dataflow search. Although GPU can accelerate DCP by leveraging the differentiable property of back-propagation, we perform the dataflow search of DCP and other baseline methods on CPU (i5-12500H) for a fair comparison. From Table 1, we see that our DCP is more efficient than other baseline search algorithms. Note that DCP separates the dataflow searching from the data collection, and the searching can be ultra-fast once the neural predictor is trained. Hence, the advantage in the time cost of DCP can be enlarged when more models and HW metrics are evaluated.

## 5.3 Per-model dataflow optimization

Compared with reconfigurable accelerators that can utilize a better dataflow for each DNN layer, DNN accelerators designed with a fixed dataflow still dominate in existing accelerators which makes it essential to optimize dataflow for the whole DNN model. And unlike prior arts, DCP can search the optimal dataflow for both DNN layers and an entire DNN model. In model dataflow optimization, we search for the optimal dataflow of three well-known DNN models based on three different metrics (e.g., EDP, latency, and energy). We further compare the performance of DCP searched dataflow with the dataflow of three well-known state-of-the-art DNN accelerators, which are NVDLA (nvd, 2018), Eyeriss (Chen et al., 2017), ShiDianNao (Du et al., 2015) and the dataflow searched by NAAS (Lin et al., 2021). NAAS is a co-design algorithm that targets the same HW platform as ours, and it also performs an experiment that only searches the dataflow in its paper. As is shown in Table 2, DCP outperforms the DNN accelerators' best performance in all evaluation metrics.

**Analysis of Searched Dataflow Design.** An example of the model dataflow searched for MobileNet-v2 is shown in Fig. 3. The searched design improves the performance of running MobileNet-V2 by leveraging its computation and storage characteristics. Mobilenet-v2 widely uses depthwise layers, and it is more efficient to explore parallelism in dimensions of input/output channels Li et al. (2022). In specific, the searched design comprises one L2 cluster and seven L1 clusters optimized for parallelization in the output channel dimension. Each L1 cluster consists of 24 PEs that perform computation parallelized over the input channel dimension and prior to performing temporal computation over the output channel dimension. Additionally, the searched dataflow also features a data buffering strategy that prioritizes buffering data in the row/column of activations dimension. This design can leverage the characteristics of depthwise convolution and weight distribution in MobileNet-V2 to minimize memory access. By contrast, NVDLA's computation in weight-stationary dataflow prioritizes parallelization primarily in the input/output channel dimension. However, it is crucial to note that the superior dimension in NVDLA's data buffering strategy is the row/column of the filter weight. This can result in the under-utilization of on-chip memory and inefficient computation in depthwise convolution-dominated operations, such as MobileNet-v2.

## 5.4 Optimizing Dataflow for Multiple Objectives.

Unlike the other baseline methods, DCP can optimize the dataflow for both single and multiple objectives as introduced in Sec 4.3. In the following experiments, we optimize three metrics in both single objective and multiple objectives optimization with the pairwise combination of the three metrics. The selected three objective HW metrics are latency, energy, and EDP, and the detailed experimental data are summarized in Table 3. As it is shown in Table 3. Although optimizing dataflow for multiple objectives cannot achieve the best performance in corresponding metrics compared with single objective optimization, it can achieve comparable performance in multiple metrics simultaneously. For example, compared with optimizing dataflow only for energy, optimizing dataflow for both energy and EDP can significantly improve the performance of latency and EDP while still having a comparable energy consumption.

## 5.5 Optimizing Dataflow for Unseen HW Settings.

In this experiment, we execute DCP in zero-shot or few-shot unseen HW settings to evaluate the generalization of DCP. Initially, the neural predictor is trained with a dataset that randomly sampled $2K$ dataflows for each DNN layer under the HW setting of Eyeriss chip (Chen et al., 2017), which is 168 PEs and 108KB on-chip memory. In zero-shot settings, we execute DCP with this original neural predictor. Then in the few-shot settings, we fine-tune the neural predictor with a dataset that randomly sampled 200 dataflows for DNN layers within the previous dataset under the corresponding unseen HW settings. The selected unseen HW settings share the same on-chip memory as the Eyeriss chip but with different PEs. As demonstrated in Fig. 6, DCP in the zero-shot setting outperforms GAMMA (best baseline method performed in Sec 5.2) in MobileNet-V2, ResNet101, and ViT, while GAMMA needs to evaluate $10K$ samples from scratch for each DNN layer. Then for the few-shot setting, DCP can search the better dataflow after fine-tuning the neural predictor.

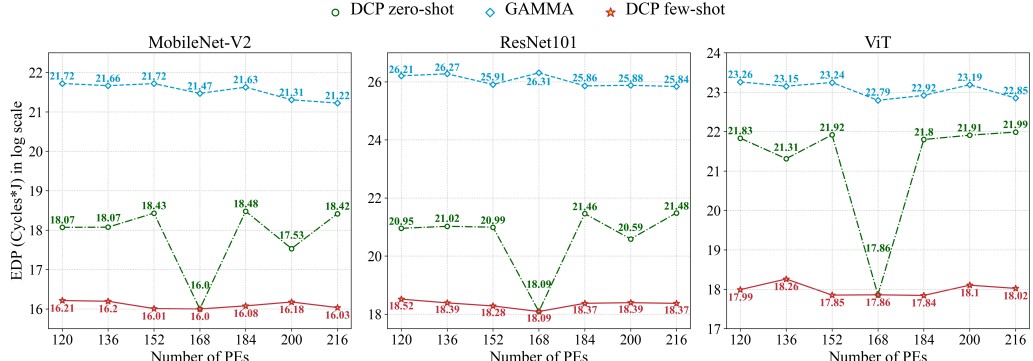

Figure 6: Performance (EDP) on three DNN models of the dataflow searched on different HW settings. Three approaches are compared in this figure. "DCP zero-shot" denotes the dataflow searched with the neural predictor only trained on the 168 PEs, and "DCP few-shot" denotes the dataflow searched with the neural predictor tuned with 0.1× samples of corresponding PEs. "GAMMA" denotes the dataflow searched by the specifically designed GA algorithm, GAMMA (Kao & Krishna, 2020).

Table 4: Performance of optimized dataflow on MobileNet-v2 for the prediction model trained with different predictor architecture and the numbers of dataflow sampled in each layer.

| Methods | Predictor Architecture | | Data Size | | | | |
|---|---|---|---|---|---|---|---|
| | NCP | DCP | 500 | 1000 | 2000 | 3000 | 4000 |
| EDP | 2.33e+09 | 8.88e+06 | 4.84e+09 | 1.97e+09 | 8.88e+06 | 8.65e+06 | 6.44e+06 |

### 5.6 Ablation Study

In this section, we further conduct some ablation experiments for the impact of two design decisions of DCP, which is inclusive of i) the dataflow sampled in each layer, and ii) the architecture of the neural predictor.

The detailed experiment results are depicted in Table 4. And our experimental analysis is performed on MobileNet-v2, utilizing the Energy-Delay Product (EDP) as the objective metric. In Table 4, we first ablate the choice of the neural predictor's network architecture. We make comparisons with the predictor architecture in NCP (Ding et al., 2021). The predictor in NCP employs Multilayer Perceptron (MLP) modules, whereas our predictor is primarily constructed using attention modules which have advantages in modeling relationships between input entries. We see that the attention-based predictor in DCP achieves better performance than the MLP-based predictor. Then, we conduct evaluations of the impact of the dataflow sampled at each layer. It can be seen that sampling 2000 dataflow codes for each layer are enough to produce good HW performance.

## 6 Conclusion

In this paper, we present a differentiable back-propagation approach, DCP, to automatically search the optimal dataflow for DNN accelerators. Dataflow is crucial for designing a DNN accelerator, but identifying the best dataflow is challenging due to its massive design space. Therefore, we propose an encoding scheme to project the dataflow and DNN layer configurations into a unified coding representation and build a comprehensive benchmark. DCP can optimize dataflow for single/multiple layers and various optimization objectives. Our results show that DCP outperforms the existing dataflow optimization methods in both optimization performance and time cost. DCP also outperforms prior arts in zero-shot or few-shot manners without time-consuming simulations, while prior arts need to re-run the simulations from scratch for every search.

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

## Appendix

In the Appendix, we provide more details of dataflow and an experiment about performing DCP in a realistic HW platform in Sec. A and Sec. B, respectively. Additionally, an ablation study for the design decision of DCP is illustrated in Sec. C.

## A    Dataflow

Section 3 of the main paper provides a brief introduction of dataflow, and more details of dataflow are introduced in this section. Dataflow is the data communication pattern within DNN accelerators between the compute and memory elements. Dataflow affects the data reuse pattern, which is critical to the throughput and energy efficiency of the accelerator.

### A.1    Motivation

Firstly, it has been shown that the energy cost of moving data exceeds the cost of computation (Horowitz, 2014). So understanding and optimizing dataflow is a critical component of DNN accelerator design as it directly determines how data is transferred between different levels of buffers in the memory hierarchy.

Secondly, there are multiple data reuse opportunities in DNN accelerators, making dataflow design necessary. Two examples of data reuse taxonomy in DNN accelerators are multicasting and reduction.

**Multicasting.** In running a DNN application, there are multiple opportunities to reuse data, like input tensor reuse and filter weight reuse, which can be leveraged by multicasting. Multicasting means reading a data point from a buffer only once and replicating it spatially or temporally. Spatial multicasting means delivering the data point to multiple spatial destinations (PEs), and it can reduce expensive memory access and data transfer to save energy and decrease latency. Temporal multicasting means delivering the data to multiple temporal destinations (same PE at different timestamps), and it has the same functionality as spatial multicasting.

**Reduction.** Reduction means accumulating partial outputs spatially or temporally to get the destined output. For example, every convolution output accumulates multiple partial outputs by multiplication input tensors and weights. Spatial reduction means collecting these partial outputs from multiple spatial destinations, while temporal reduction means gathering them from multiple temporal destinations.

### A.2    Expression

Specifically, we use three factors to describe a dataflow: parallelism, computation order, and partitioning size.

**Parallelism.** Parallelism is about allocating the computation workload into the multiple hardware processing units and letting these processing units can perform the computation simultaneously. For DNN accelerators, the processing units are PEs, and it usually achieves parallelism via unrolling the loop nest of convolutional layers spatially.

**Computation order.** Computation order is the temporal order of the dimensions to be performed in the multiply dimensional computation task. Different computation orders can exploit different data movement patterns and reuse opportunities.

**Partitioning size.** As there are many parameters to the DNN application and the buffer of DNN accelerators is limited, accelerators with multi-level memory hierarchy will partition these parameters and allocate them into the buffer. In this process, partitioning size will determine the data size of each tensor (input/output/weight) that needs to be present within the accelerator's buffer at each timestamp.

### A.3    Example

A simple 1-D convolution and its loop expression are shown in Figure 7. The loop shown in Figure 7b is time-consuming as it only performs one multiplication and one add operation at each time step. Suppose we apply

the parallelism to this 1-D Conv by unrolling the loop and using three processing elements simultaneously to perform the computation task. In that case, the 1-D Conv will be conducted as shown in Figure 7c.

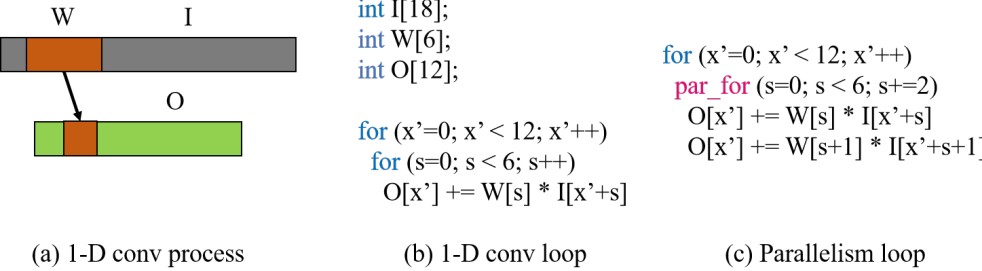

(a) 1-D conv process      (b) 1-D conv loop      (c) Parallelism loop

Figure 7: The computation process and loop expression of 1-D Conv. (a) An overview of the 1-D Conv process and W, I, and O represent the filter weight, input, and output activations, respectively. (b) The loop expression of the 1-D Conv process. (c) The loop expression of performing parallelism on 1-D Conv using three PEs.

Then for the effect of computation order, it is illustrated in Figure 8. Stationary means the corresponding data stay the longest in the accelerator's buffer. As it is shown in Figure 8, "tmp" refers to one data in the accelerator's buffer, and different computation orders may result in different numbers of memory access for each tensor.

Output stationary

```
for (x'=0; x' < 12; x'++)
  tmp = 0
  for (s=0; s < 6; s++)
    tmp += W[s] * I[x'+s]
  O[x'] = tmp
```

Output: 12 read and 12 write
Weight: 6*12 read
Input: 6*12 read

Input stationary

```
for (x=0; x < 18; x++)
  tmp = I[x]
  for (s=0; s < 6; s++)
    O[x-s] += W[s] * tmp
```

Output: 6*12 read and 6*12 write
Weight: 6*12 read
Input: 18 read

Weight stationary

```
for (s=0; s < 6; s++)
  tmp = W[s]
  for (x'=0; x' < 12; x'++)
    O[x'] += tmp * I[x'+s]
```

Output: 6*12 read and 6*12 write
Weight: 6 read
Input: 6*12 read

Figure 8: Memory access on three different computation order patterns of performing 1-D Conv.

As DNN accelerators have a multi-level memory hierarchy and a limited size buffer. Therefore, the data used in DNN applications may need to be partitioned before being transferred into the DNN accelerators and performing the computation. An example of partitioning the data used in 1-D Conv into different memory levels within the DNN accelerator is shown in Figure 9. In Figure 9, the data of input activation has been partitioned into smaller tiles and transferred into different memory levels.

## B Tensorlib

As it is mentioned in Section 5, we can further generate synthetic circuits for the dataflow optimized by DCP, unlike prior arts only based on the performance simulators. To do this, we leverage Tensorlib (Jia et al., 2021), a spatial accelerator generation framework. More details are introduced in this section. We choose Tensorlib because it is open-sourced and also provides an elegant way to model the computation of accelerators, which is the Space-Time-Transformation (STT) matrix.

### B.1 Design space

To generate the dataflow satisfying the constraint of Tensorlib, we limit the search space of DCP with the design space of Tensorlib. The design space Tensorlib is mainly inclusive of three parts, which are buswidth, Space-Time Transformation (STT) matrix, and intrinsic size. Between the spatial accelerator and the memory, the is a data bus to transfer the data, and buswidth defines the width of this data bus in units of bits. Then, the STT matrix is a mean of expression to represent the hardware dataflow. And intrinsic size is used to

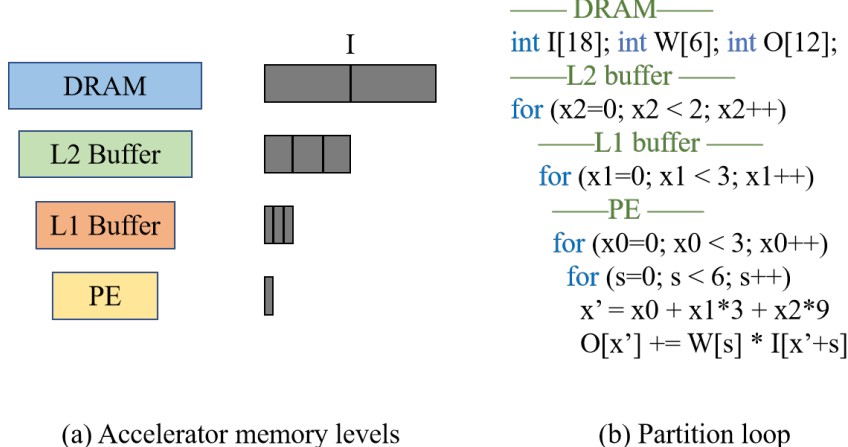

(a) Accelerator memory levels         (b) Partition loop

Figure 9: Data partitioning example. (a) An overview of the accelerator's memory level and input activation (I) data size in each level. (b) The loop expression of partitioning input activation on 1-D Conv.

describe the size of the loop nest workload. As the spatial accelerator generated by Tensorlib targets the computation workload that can be described in a perfect nested loop. In executing such a computation workload, the PE array can be viewed as a hypercube, and the execution of hardware can be identified as a space vector and a time scalar indicating where and when the computation task is placed. In detail, for every point in the loop nest, the STT matrix can transform into the space-time vector in hardware execution using a matrix multiplication operation. Then this space-time vector can map loop instances to hardware spatially (coordinates in the PE array) and temporally (timestamp of execution).

## B.2   Example

A detailed explanation of how we can get the space-time vector introduced in Sec B.1 is described in this section. Given a loop iteration in the loop nest $x = [i, j, ...]^T$ and an STT matrix, the execution space and time can be calculated as $[p, t]^T = STT \cdot x$, where space vector $p$ means the PE coordinates inside the PE array and time scalar t means the time step of execution. A simple example that targets the computation workload of 2D matrix multiplication is shown in Figure 10. What's more, we can get the upper bound and lower bound of PE coordinates and timestamp of execution by multiplying the intrinsic size and zero vector with the STT matrix and further calculating the size of the PE array and spatial accelerator's memory.

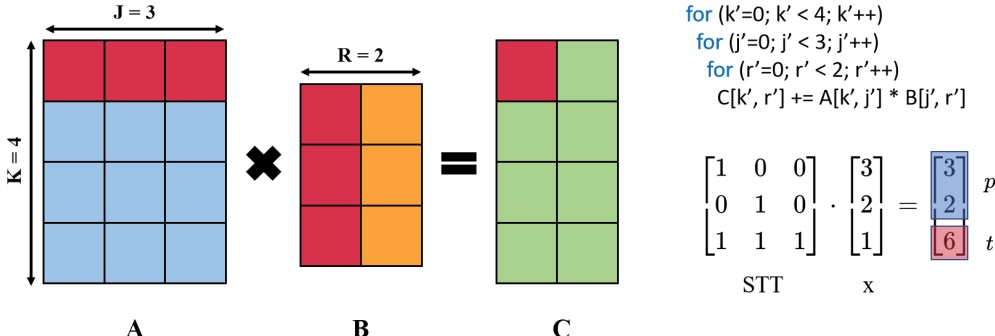

Figure 10: Space-Time Transformation example with 2D matrix multiplication.

## B.3   Experiment

To show the generalization of DCP, we build a benchmark of synthetic dataflow and DNN layer based on Tensorlib and then follow the workflow of DCP to search the optimal dataflow. After that, we generate the

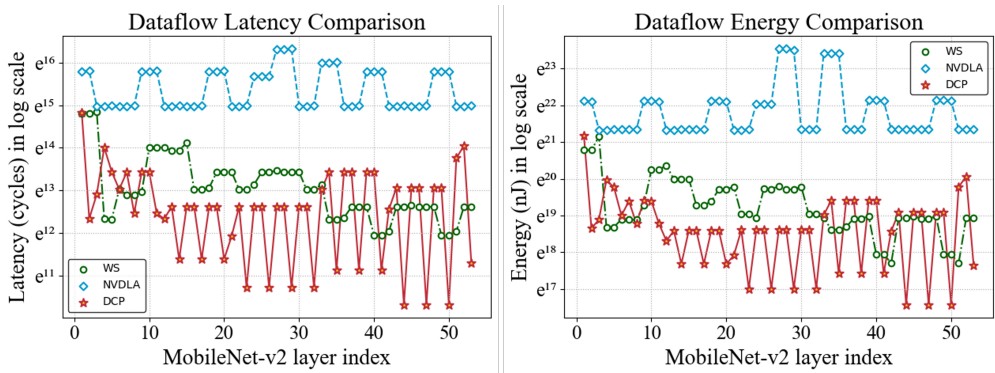

Figure 11: Layer-wise performance of DCP optimized dataflow and example dataflow provided by Tensorlib in the latency and energy consumption of MobileNet-v2. "WS" denotes the exemplary weight-stationary dataflow provided in Tensorlib. "NVDLA" denotes the dataflow of NVDLA (nvd, 2018) accelerate. "DCP" denotes the dataflow searched by DCP.

synthetic circuit for the searched dataflow and perform a cycle-accurate simulation to show its performance. To build the benchmark for Tensorlib, We rebuild a new dataset that shares the same collection of DNN layers with the previous dataset but with $2K$ dataflows randomly sampled in the design space of Tensorlib. As the accelerator generated by Tensorlib has a fixed dataflow, we compare the performance of DCP-optimized dataflow with other fixed dataflow accelerators for a fair comparison. Fig. 11 compares the achieved layer-wise performance of the accelerator generated by the DCP-optimized dataflow and other exemplary dataflow provided in TensorLib. It is shown that a significant improvement is achieved by the dataflow optimized by DCP in both latency and energy consumption of MobileNet-V2. The exemplary dataflow provided in TensorLib is a kind of weight-stationary (WS) dataflow. It's worth noting that although many accelerators use WS dataflow, each of them may have some differences. For example, Google's TPU (Jouppi et al., 2017) and Intel's Gaudi (gau, 2020) are both custom-designed accelerators that use WS dataflow, but they have different architectures. TPU uses a unified on-chip memory to hold data, while Gaudi has a more fine-grained distributed memory architecture.

