# OpenReview forum: "DCP: Learning Accelerator Dataflow for  Neural Network via Propagation"
_TMLR — Rejected by TMLR_

### Review · Reviewer_ozmZ · 2023-07-08

**Summary Of Contributions:**

This paper presents a framework to automatically learn the optimal dataflow for hardware accelerators executing deep CNNs. The authors abstract each CNN layer into 7 key parameters and the hardware architecture as containing a memory hierarchy and an array of PEs. The authors further constrain the kind of operations (e.g., partitioning, re-ordering, etc) corresponding to each dataflow configuration. Based on such abstraction, the authors are able to formalize a coding scheme to represent possible dataflow and such coding is used as the input to the neural network predictor. The training of the predictor is based on a manually constructed benchmark via simulation. The inference of the predictor directly updates the coding so that the dataflow can be optimized under various target metrics. Experiments show that the automatically learnt dataflow achieves significantly better performance than manual design.

**Audience:**

Yes

**Broader Impact Concerns:**

None that I am aware of.

**Claims And Evidence:**

No

**Requested Changes:**

* Please specify what kind of hardware architectures, CNN algorithms and partitioning / unrolling schemes can be supported by DCP.
* Please describe how to perform hardware feasibility check for a new type of hardware.
* Please include a discussion on accelerators for fast convolution algorithms.
* Please include comparison with more relevant / recent baselines.

**Strengths And Weaknesses:**

Strengths
+ It is interesting to have a joint abstraction on the dataflow, CNN layer parameters and the target hardware. Such an abstraction enables a coding scheme which generates the predictor neural network's inputs.
+ The predictor is able to optimize multiple hardware objectives. This potentially makes the design more general and impactful.
+ The proposed design achieves significant performance improvement compared with baselines.

Weaknesses
- The proposed framework may only be applicable to some specific types of CNN models, execution algorithms, dataflow schemes and hardware architectures. For example, a lot of the hardware assumptions seem to be heavily based on a specific hardware chip, Eyeriss. For ASICs or FPGAs, the PEs and on-chip buffers can be flexibly organized, in a form not necessarily captured by Fig 2. Ideally, the parallelization can also be performed in a multi-level fashion but the encoding of DCP seems to only support a single level of unrolling. The hardware feasibility check for the generated dataflow is also following the constraints imposed by Eyeriss, which is restrictive.
- For convolution layers, there is a rich set of literature focusing on hardware acceleration of fast convolution algorithms, such as frequency domain convolution [a] or Winograd convolution [b]. Accelerators for fast convolution algorithms may achieve much better performance than the original spatial convolution accelerators (as considered by this paper). Therefore, it is unclear whether this paper's contributions are significant if the proposed DCP does not incorporate fast convolution algorithms.
- The paper is motivated by a huge design space consisting of all possible data flows. However, the challenges due to the size of the design space may be exaggerated. The paper shows an example of an O(10^36) design space. However, this space can be dramatically reduced by some simple condition checks and filtering. In addition, many works on ASIC and FPGA accelerators build accurate performance models that enable fast design space exploration. Since the CNN data has a fixed shape, the performance model should give a quite accurate estimation without the need of expensive simulation.
- The baselines in experiments are quite old. There are many more recent works accelerating CNNs by customized dataflow.

References:
[a] Zeng et al., A Framework for Generating High Throughput CNN Implementations on FPGAs. In ACM/FPGA 2018
[b] Lavin et al., Fast Algorithms for Convolutional Neural Networks. In CVPR 2016

---

### Review · Reviewer_o2wX · 2023-07-10

**Summary Of Contributions:**

The paper proposes Dataflow Code Propagation (DCP), a method for automatically optimizing the dataflow in deep neural network layers, improving performance and efficiency. Specifically, DCP translates hardware dataflow configurations into a coding space for optimization. Additionally, it can adapt to new hardware configurations via zero-shot or few-shot learning, outperforming existing methods such as GAMMA. Experimental results on MobileNetv2, ResNet, and VIT models demonstrate DCP's improvements over baselines.

**Audience:**

Yes

**Broader Impact Concerns:**

No concern on the ethical implications of the work

**Claims And Evidence:**

Yes

**Requested Changes:**

Please see the "Weaknesses/Questions" part.

**Strengths And Weaknesses:**

# Strengths
1. Comprehensive summary of prior works: the prior work section covers general DNN accelerators, design space of DNN accelerators, simulators of DNN accelerators, and co-design of accelerators. The provided preliminary about the DNN accelerator is concise and clear.
2. New pipeline to get more efficient dataflows: while most existing works focus on searching for the best dataflows given the hardware resources or manually tuning it, this work proposes a new pipeline based on gradient back-propagation. Specifically, a DNN hardware efficiency metric predictor is first trained, and then the weights are fixed to update the input dataflow embeddings with the goal of getting the dataflow that can make the target metric smallest.
3. Impressive results: the optimized dataflow achieved much better (e.g., 100 times) efficiency performance as compared to baselines.

# Weaknesses/Questions
1. The hardware efficiency metric used for training is obtained by simulators. How to justify that they can provide numbers that are very similar to the real devices?
2. NVDLA and Eyeriss are highly optimized for CNN workload. How did the authors change those accelerators for ViT workload? Is it fair to compare with NVDLA and Eyeriss when targeting ViT workload? It seems that the proposed method can achieve higher improvement on the ViT workload.
3. Although the proposed method is based on gradient backpropagation, it requires the background knowledge of DNN hardware accelerator. Thus, I am not sure whether TMLR is the best venue for it. Works focusing on applying deep learning to architecture are highly appreciated in the computer architecture area (e.g., https://arxiv.org/abs/2209.00188 best paper of MIRCO'22 about applying machine learning to Off-Chip Load Prediction).
4. Transferability to more complex DNNs: with DNNs are designed to be more complex (i.e., mixing different workloads and more inter-layer connections), will the proposed method still be able to provide better optimized dataflow than manually designed ones or other domain-specific dataflow optimization workflow. For example, there are some works focusing on mixing CNN and Transformer within the same layer (https://arxiv.org/abs/2004.11886). Meanwhile, some works find that inter-layer scheduling is also an important factor in the whole model's hardware efficiency (https://dl.acm.org/doi/abs/10.1145/3579371.3589048, https://arxiv.org/abs/2011.01302). Considering that this work only considers the hardware efficiency metric for each layer, how to change to consider inter-layer scheduling and the scalability to DNN with many inter-layer connections (e.g., https://arxiv.org/abs/1806.09055) may be a problem.

---

### Review · Reviewer_5a5r · 2023-07-19

**Summary Of Contributions:**

This work introduces Dataflow Code Propagation (DCP), an approach to derive optimal dataflows for DNN layer, that take into account parallelization of the operations across processing units, partitioning over multiple memory levels, and order of the computations. The core idea is to train an attention-based network to predict relevant metrics (latency, energy, or their combination) based on the input dataflow encoding of a DNN layer, then iteratively obtain the optimal encoding subject to constraints and objectives. Results are evaluated on 3 separate DNN models (MobileNet-v2, ResNet101, ViT) and compared to several competing dataflow optimization frameworks.


**Audience:**

Yes

**Broader Impact Concerns:**

No specific concerns

**Claims And Evidence:**

Yes

**Requested Changes:**

- Add comparison with Ding et al. to Section 2 "Related Work"
- I would strongly recommend a revision of the manuscript, possibly with the help of a native speaker, as several grammatically incorrect expressions can be found throughout it

**Strengths And Weaknesses:**

Strengths:
- The topic of DNN runtime and energy consumption optimization, subject to HW constraints, is highly relevant
- The text is well structured, clear, and easy to follow (despite grammatical errors, see Requested Changes)
- Extensive comparison of results against several other dataflow optimization techniques and frameworks; reported results are quite impressive, with DCP outperforming competing methods by one order of magnitude of more across the evaluated metrics
- The methodology also appears to be advantageous in terms of optimization runtime

Requested clarifications and other comments:
- To train the predictor, 2K DNN layers are randomly selected, and 2K random dataflows are samples per layer, requiring a total of 4M evaluations. Is this a one-time cost? How expensive is this operation? Is this cost included in the time cost estimates of Table 1?
- When iterating on a frozen predictor to optimize the dataflow encoding (Fig. 4 and Section 4.3), is the newly obtained encoding used as input to the following iteration, at each step? If so, Fig. 4 could be modified to reflect this visually and the text could clarify this.
- The work leverages the work by Ding et al. "Learning Versatile Neural Architectures by Propagating Network Codes" ICLR 2022, which introduced the use of design space search with codes and a trained neural predictor for Neural Architecture Search (NAS). While this work is referenced in Section 5.2 "Ablation Studies", it should be given more prominence as several core concepts overlap (herein applied on the distinct topic of dataflow optimization, and using an attention-based network instead of an MLP).

---

### Decision · Action_Editors · 2023-08-21

**Recommendation:** Reject

**Comment:**

All reviewers agreed that this submission has value. They unanimously agreed that the topic is of interest to the TMLR audience (although one reviewer suggested that a computer architecture venue might be an even better place to submit this work), that the reported results were extremely impressive, and that the text was generally well written, modulo some issues with incorrect English usage. The reviewers did, however, raise a number of points that, if addressed, would greatly improve the paper. Among these are the question of how specific the proposed DCP method is to the Eyeriss accelerator, whether or not the size of the search space is overstated in the paper because it can in practice be reduced via simple condition checks, the need for comparison against more current baselines, the extent to which the simulator results will transfer to actual hardware, whether the proposed method will work with more complicated network architectures, and the need to more clearly relate the work in the submission to prior work by Ding et al. in ICLR 2022. Regrettably, the authors did not take advantage of the response and discussion period, which is an integral part of the TMLR review process, to address these issues.

**Audience:**

Yes - the reviewers are unanimous in saying that the paper would be of interest to the TMLR audience.

**Claims And Evidence:**

No - authors failed to address concerns about the generalizability of the proposed DCP method to different accelerator designs and different neural network architectures, about the relative age of the baselines used in the paper, and whether or not simple condition checks would reduce the size of the search space.

**Resubmission Of Major Revision:**

The authors may consider submitting a major revision at a later time.